# Generalized Cross-Correlation Strain Demodulation Method Based on Local Similar Spectral Scanning

**DOI:** 10.3390/s22145378

**Published:** 2022-07-19

**Authors:** Yuqi Tian, Jiwen Cui, Zaibin Xu, Jiubin Tan

**Affiliations:** 1Center of Ultra-Precision Optoelectronic Instrument, Harbin Institute of Technology, Harbin 150080, China; 19b901006@stu.hit.edu.cn (Y.T.); 20b901004@stu.hit.edu.cn (Z.X.); jbtan@hit.edu.cn (J.T.); 2Key Lab of Ultra-Precision Intelligent Instrumentation, Harbin Institute of Technology, Ministry of Industryand Information Technology, Harbin 150080, China

**Keywords:** optical fiber measurement technology, generalized cross-correlation, local similar spectral scanning

## Abstract

Optical fiber measurement technology is widely used in the strength testing of buildings, the health testing of industrial equipment, and the minimally invasive surgery of modern medical treatment due to its characteristics of free calibration, high precision, and small size. This paper presents an algorithm that can improve the range and stability of strain measurements in order to solve the problems of the small range and measurement failure of optical fiber strain sensors based on optical frequency-domain reflectometry (OFDR). Firstly, a Rayleigh scattering model based on the refractive index perturbation of an optical fiber is proposed to study the characteristics of Rayleigh scattering and to guide the strain demodulation algorithm based on the spectral shift. Secondly, a local similar scanning method that can maintain a high similarity by monitoring local Rayleigh scattering signals (LSs) before and after strain is proposed. Thirdly, a generalized cross-correlation algorithm is proposed to detect spectral offset, solving the problem of demodulation failure in the case of a Rayleigh scattering signal with a low signal-to-noise ratio. Experiments show that the proposed method still has high stability when the spatial resolution is 3 mm. The measurement precision is 6.2 με, which proves that the multi-peaks or pseudo-peaks of the traditional algorithm in the case of a large strain, the high spatial resolution, and the poor signal-to-noise ratio are solved, and the stability of the strain measurement process is improved.

## 1. Introduction

Fiber-optic strain-sensing technology has been widely used in building and industrial health monitoring, puncture needles, and minimally invasive surgical equipment [1,2,3,4,5]. Rayleigh-spectrum-addressing technology based on optical frequency-domain reflectometry has attracted extensive attention because of its high spatial resolution and measurement accuracy [6]. In 1998, Froggatt calculated the Rayleigh spectral shift to determine the axial strain on an optical fiber with cross-correlation, and this method has been widely used since it was proposed [7]. In 2012, Zhou compared the strain measurement quality of an OFDR system based on Rayleigh scattering with different spatial resolutions [8]. In 2016, a strain measurement system with a spatial resolution of 4.5 cm and a minimum strain of 3 με was realized by Ding [9].

However, this method ignores the differences in the spectral characteristics of an optical fiber under load, leading to spectral degradation in the process of large-range measurements and affecting the final measurement results. In 2017, Gan demodulated the strain of an optical fiber using the phase information of the Rayleigh signal in the distance domain. The strain detection range was only a few micro-strains, and the spatial resolution was low due to the influence of the phase amplitude [10]. In 2017, Kim realized the measurement range of 1000 με using the Fourier transform processing of a bidirectional-offset Rayleigh scattering spectrum [11]. In 2018, Heinze extended the strain measurement range by replacing the fixed Rayleigh spectrum with a dynamic Rayleigh spectrum, superpositioning the previous small strains into the final large-strain measurement results [12]. However, the measurement range of this method is still small, and the final strain measurement error is limited by the cumulative error of several iterations. In addition, the low signal-to-noise ratio (SNR) of the Rayleigh scattering signal results in the failure of strain measurement. In order to improve strain resolution and accuracy, Loranger increased backscattering by irradiating the fiber under text (FUT) with UV light to increase the signal-to-noise ratio of the detection signal [13]. Kreger used a plastic optical fiber to prove that an optical fiber with a large refractive index fluctuation has stronger backward Rayleigh scattering, but the plastic optical fiber was more sensitive to the environment, resulting in poor system stability [14]. In 2017, Yan irradiated the fiber under text with an ultra-narrow linewidth laser to increase the strength of the Rayleigh scattering signal, thus increasing the effect of cross-correlation calculations [15]. Nonetheless, the above methods can effectively improve the signal-to-noise ratio of the signal, but the spectral degradation in large-strain demodulation cannot be ignored because the processing of the optical fiber destroys the microstructure of the optical fiber.

In summary, the current distributed strain measurement system based on optical frequency-domain reflectometry mainly has the following problems: (1) Rayleigh scattering is only considered to be a weak grating spectrum in previous research, and it cannot explain the problem of spectral degradation when an axial strain occurs in the fiber. (2) The quality of the cross-correlation results decrease in large-strain measurements due to spectral degradation, thus affecting strain demodulation. (3) Cross-correlation is limited by the signal-to-noise ratio (SNR) of the detection signal, resulting in the problem of multi-peaks or pseudo-peaks, which makes it impossible to effectively extract the spectral offset. In order to solve these problems, the main work of this paper is as follows: (1) A Rayleigh scattering model is proposed based on the refractive index perturbation of an optical fiber to study the characteristics of Rayleigh scattering spectra and to guide the strain demodulation algorithm based on the spectral shift (2) A spectral shift detection method based on local similar spectral scanning is proposed to solve the problem of spectral similarity reduction caused by spectral degradation due to large strains. (3) A strain demodulation algorithm based on a generalized cross-correlation is proposed to solve the problem of cross-correlation operation failure caused by the low signal-to-noise ratio of the Rayleigh scattering signal. Experiments show that the proposed method greatly improves the stability of optical fibers under low signal-to-noise ratios and improves large-strain measurement conditions on the premise of ensuring spatial resolution.

## 2. Theory

### 2.1. Rayleigh Scattering in Optical Fibers

Rayleigh scattering is caused by the inhomogeneity of the medium. Rayleigh scattering in optical fibers can be derived from the longitudinal mode-coupling theory, which is based on the perturbation solution. When the light shoots into an optical fiber, forward and backward beam coupling is generated due to changes in the refractive index, and the results are directly given as follows:(1)$$\{\begin{array}{c} \frac{dc_{u}^{+}}{dz}=\sum_{v}\left\{\kappa_{uv}^{+}c_{v}^{+}\exp[i(\beta_{u}-\beta_{v})z]+\kappa_{uv}^{-}c_{v}^{+}\exp[i(\beta_{u}+\beta_{v})z]\right\}\\ \frac{dc_{u}^{-}}{dz}=-\sum_{v}\left\{\kappa_{uv}^{-}c_{v}^{+}\exp[-i(\beta_{u}+\beta_{v})z]+\kappa_{uv}^{+}c_{v}^{-}\exp[-i(\beta_{u}-\beta_{v})z]\right\} \end{array}$$
(2)$$\{\begin{array}{c} \kappa_{uv}^{+}=\frac{1}{2}(\kappa_{uv}^{\left(1\right)}+\kappa_{uv}^{\left(2\right)})\\ \kappa_{uv}^{-}=\frac{1}{2}(\kappa_{uv}^{\left(1\right)}-\kappa_{uv}^{\left(2\right)}) \end{array}$$
where $c_{u}^{+}$ and $c_{u}^{-}$ are the amplitudes of the forward and backward propagation of *u* mode, respectively; $c_{v}^{+}$ and $c_{v}^{-}$ are the amplitudes of forward and backward propagation of *v* mode, respectively. $\beta_{u}$ and $\beta_{v}$ are the propagation constants of *u* and *v* modes, respectively. $\kappa_{uv}^{+}$ and $\kappa_{uv}^{-}$ are the forward coupling coefficient and reverse coupling coefficient, respectively. $\kappa_{uv}^{\left(1\right)}$ and $\kappa_{uv}^{\left(2\right)}$ are the scalar product of the mode transverse electric field and the mode longitudinal electric field, respectively, which can be expressed numerically as follows:(3)$$\begin{array}{l} \kappa_{uv}^{\left(1\right)}=\frac{\omega \varepsilon_{0}}{2i}\iint \left(n^{2}-n_{0}^{2}\right)e_{ut}^{*}e_{vt}dxdy\\ \kappa_{uv}^{\left(2\right)}=\frac{\omega \varepsilon_{0}}{2i}\iint \frac{n_{0}^{2}}{n^{2}}(n^{2}-n_{0}^{2})e_{uz}^{*}e_{vz}dxdy \end{array}$$
where *n* and *n*_0_ are the actual and ideal refractive index of the optical fiber, respectively; *ω* is the frequency of the light wave; and *ε*_0_ is the dielectric constant of the vacuum. **e***_vt_* and **e***_vz_* are the transverse and longitudinal electric field components of *v* mode, respectively. $e_{ut}^{*}$ and $e_{uz}^{*}$ are the complex conjugation of the transverse electric field component and the longitudinal electric field component of *u* mode, respectively.

According to the completeness of the modes, the summation process in Formula (1) indicates that the total optical wave field in the fiber is the superposition of each ideal optical waveguide mode. Therefore, when the refractive index of the fiber is not uniform (that is, when there is perturbation), the incident light with mode *v* will produce forward- and backward-propagating ***u***-mode light. This paper mainly studies backward Rayleigh scattering based on a single-mode fiber, so only the second equation in Formula (1) and *u* = *v* = 1 in a single-mode fiber are considered. In addition, the scattering intensity is much less than the incident light intensity $\left|c_{0}\right|\gg \left|c^{-}\right|$. Thus, Formula (1) can be simplified to
(4)$$\frac{dc^{-}}{dz}=\kappa^{-}c_{0}\exp(-2i\beta z)$$

When the light wave propagates in the optical fiber, the longitudinal component of the electric field is much smaller than the transverse component, so the second equation in Formula (3) can be ignored. Assuming that the polarization direction of the transverse electric field component is *x*, the inverse coupling coefficient is obtained as follows:(5)$$\kappa^{-}=\frac{1}{2}(\kappa_{11}^{\left(1\right)}-\kappa_{11}^{\left(2\right)})=\frac{\omega \varepsilon_{0}}{4i}\iint \left(n^{2}-n_{0}^{2}\right)e_{x}^{*}e_{x}dxdy$$

Since the transverse perturbation of the refractive index has little effect on Rayleigh backscattering, the longitudinal perturbation of the refractive index is considered; that is, the refractive index is a function of the fiber position z, so Formula (5) can be written as
(6)$$\kappa^{-}=\frac{\omega \varepsilon_{0}Z_{0}}{4i\cdot n_{0}}(n^{2}-n_{0}^{2})\iint e_{x}^{*}h_{y}dxdy$$
where *Z*_0_ is the wave impedance, and *Z*_0_ = $\sqrt{\mu_{0}/\varepsilon_{0}}$. **h***_y_* is the magnetic field in the *y* direction. According to the orthogonal normalization relationship of the ideal waveguide mode, Formula (6) can be derived as
(7)$$\kappa^{-}=\frac{\omega \varepsilon_{0}Z_{0}}{4i\cdot n_{0}}(n^{2}-n_{0}^{2})$$

Formula (7) can be substituted into Formula (4), and both sides of the equation can be integrated with respect to position *z* to obtain
(8)$$c^{-}=\frac{k_{0}c_{0}}{4i\cdot n_{0}}\int \left(n^{2}-n_{{}^{0}}^{2}\right)\exp(-2i\beta z)dz$$

According to Formula (8), the relationship between the Rayleigh scattering amplitude and the refractive index change in the fiber is obtained. The Fourier transform of the refractive index fluctuation on the right side of the equation reflects the distribution characteristics of the longitudinal refractive index of the optical fiber. That is, the distribution of the Rayleigh scattering amplitude along the fiber is unique, and this is equivalent to the “fingerprint” of the fiber, called the Rayleigh scattering spectrum. Figure 1 shows the simulation of the Rayleigh scattering signal of the optical fiber, where the detection bandwidth is 1540–1545 nm, the total length of the optical fiber is 0.8 m, and the spatial resolution is 160 μm. Figure 1a shows the Rayleigh scattering signal in the frequency domain, and Figure 1b shows the Rayleigh scattering spectrum in the distance domain. When the fiber is stretched or compressed, the refractive index distribution of the fiber moves, leading to a corresponding shift in the Rayleigh scattering signal in the frequency domain. Thus, the strain value can be demodulated by calculating the shift.

### 2.2. Local Similar Scanning Algorithm

The traditional fiber strain demodulation system obtains the Rayleigh scattering signal before and after the strain of the fiber under text (FUT) by firstly using optical frequency-domain reflectometry technology. When the sensing fiber is in the strain-free state, the Rayleigh scattering signal in the frequency domain at each position on the fiber does not change and remains constant, and the signal at this moment is recorded as the reference Rayleigh scattering signal. When the axial strain of the sensing fiber occurs at a certain position, the Rayleigh scattering signal shifts in the frequency domain, which is proportional to the strain. The Rayleigh scattering signal is recorded at this moment as the measured Rayleigh scattering signal. The offset can be calculated by performing the cross-correlation operation of the reference and measured Rayleigh scattering signals, and the strain can be demodulated.

The essence of the traditional method is to perform the cross-correlation operation between the reference and measured signals in the whole swept (WS) frequency range. However, when the strain occurs in the sensing fiber, new components are introduced, which reduces the similarity of the Rayleigh scattering signals, thus greatly impacting the cross-correlation calculation results and affecting demodulation, as shown in Figure 2. When the tuning range is 20 nm, the sensing fiber has an axial strain of 3000 με, that is, a 3.8 nm shift, which will cause a serious degradation of the cross-correlation results. Therefore, this paper proposes a strain demodulation algorithm based on local similar scanning to solve the problem of not being able to effectively demodulate the strain due to a large strain or a high spatial resolution.

The strain demodulation algorithm based on local position scanning is described in detail below in combination with Figure 3.

(1)The tuning range of the incident light source is kept consistent, and the Rayleigh signals in the frequency domain under the strain-free and strain states of the sensing fiber are collected as the reference Rayleigh signal and the measured Rayleigh signal, respectively.(2)The reference Rayleigh signal and the measured Rayleigh signal in the frequency domain are converted into the distance domain using Fourier transform, and they are denoted as the reference Rayleigh spectrum and the measured Rayleigh spectrum, respectively.(3)The reference Rayleigh spectrum and the measured Rayleigh spectrum are intercepted by a rectangular window with length *N*.(4)The reference Rayleigh spectrum and the measured Rayleigh spectrum are supplemented with zero to number *K*, and then they are converted to the frequency domain using inverse Fourier transform to obtain the reference whole Rayleigh scattering signal (RWS) and the measured whole Rayleigh scattering signal (MWS) at a certain location. It should be noted that the bandwidth of the signal is the tuning range of the light source.(5)The rectangular window with length *A* is used to traverse the measured Rayleigh scattering signals in the whole tuning range. The signals of each intercepted region are recorded as local measured Rayleigh signals (LMSs), and the positions are represented by *p*.(6)At the center of the reference whole Rayleigh scattering signal, a rectangular window with a length of *A* is intercepted and recorded as the local reference Rayleigh signal (LRS). The correlation coefficients between the local reference Rayleigh signal and the local measured Rayleigh signal in each region are calculated step by step. The position with the greatest similarity is the shift caused by strain. When *p* = 0, the sensing fiber has no strain, and −*D* < *p* < *D* (*D* represents the boundary of the search region in the optical frequency domain).(7)The strain of the sensing fiber is calculated according to Formula (9):

(9)$$\varepsilon =\varepsilon_{res}\cdot \Delta s\cdot R_{spec}$$
where *ε_res_* represents the conversion coefficient between the strain and optical frequency, and the unit is με/GHz. Δ*s* is the minimum search step in the frequency domain, in GHz. According to step 4, the signals are supplemented by zero to point *K*, so Δ*s* = Δ*v*/*K. R_spec_* represents the signals shift.

It can be seen that the essence of the above demodulation algorithm is to cross-correlate the local measured Rayleigh signal and the local reference Rayleigh signal at the optimal position of similarity. This can effectively solve the problem of the poor cross-correlation results caused by a large strain or a high spatial resolution. The spatial resolution of the system is
(10)$$\Delta Z_{sys}=N\cdot \Delta z=N\cdot \frac{c}{2n\Delta v}$$
where *N* is the width of the sensing unit. The spatial resolution of the system is still only determined by the tuning range of the incident light and *N*. That is, in the proposed strain demodulation algorithm, increasing the width of the rectangular window (*N*) by reducing the spatial resolution is not required to increase the number of points of the cross-correlation operation. For the strain solution, the maximum measurement range can be expressed by Formula (11):(11)$$\varepsilon_{range}=\pm \frac{(\Delta \lambda_{w}-\Delta \lambda_{m})/2}{\lambda_{c}(1-P_{e})}=\pm \frac{(\Delta v_{w}-\Delta v_{m})/2}{v_{c}(1-P_{e})}$$
where ± represents the tensile or compressive strain of the sensing fiber. Δ*λ_m_* and Δ*v_m_* represent the length of the local Rayleigh signal selected in the optical frequency domain. Δ*λ_w_* and Δ*v_w_* represent the total tuning range of the light source. *λ_c_* and *v_c_* represent the central wavelength and the central optical frequency, respectively; *P_e_* is the elastic optical coefficient of the sensing fiber.

### 2.3. Generalized Cross-Correlation Algorithm

According to the above analysis, the strain demodulation algorithm based on local similar scanning can demodulate the corresponding strain by solving the optimal solution of the correlation coefficient between the reference Rayleigh scattering signal and the measured Rayleigh scattering signal, which greatly enhances the performance of the system. However, in practical engineering applications, the signal-to-noise ratio of local Rayleigh scattering signals (LSs) is small due to Fresnel reflection and other factors. In this case, the results of the cross-correlation operation cannot meet the requirements of practical applications. In order to solve the above problems, a correlation coefficient calculation algorithm based on a generalized cross-correlation is proposed in this paper, and it has better results because of its high robustness.

LRS and LMS can be expressed as *s_r_*(*n*) and *s_m_*(*n*), which are the superposition of the ideal signal *s*(*n*) and the random noise *n*_1_(*n*) and *n*_2_(*n*) of the system, respectively. Then, the signal model can be expressed using Formula (12):(12)$$\begin{array}{l} s_{r}\left(n\right)=s\left(n\right)+n_{1}(n)\\ s_{m}\left(n\right)=s\left(n-k\right)+n_{2}(n) \end{array}$$
where *k* is the shift of the signal. Then, the cross-correlation function between the reference Rayleigh scattering signal and the measured Rayleigh scattering signal is
(13)$$R_{21}=E\left[s_{r}(n)s_{m}(n)\right]=R_{ss}(n-k)+R_{sn_{2}}+R_{n_{1}s}+R_{n_{1}n_{2}}$$

According to Formula (13), the cross-correlation result is the superposition of the correlation coefficients of the ideal reference signal and the ideal measurement signal, the ideal signal and random noise *n*_1_(*n*), the ideal measurement signal and random noise *n*_2_(*n*), and random noises *n*_1_(*n*) and *n*_2_(*n*). In the ideal case, *s*(*n*), *n*_1_(*n*), and *n*_2_(*n*) are independent of each other; that is, *R_sn_*_2_, *R_n_*_1*s*_*,* and *R*_n1n2_ are equal to zero, and then Formula (13) only retains the first term *R*_ss(*n*-*k*)_. However, in practical engineering applications, *R_sn_*_2_, *R_n_*_1*s*_, and *R*_n1n2_ are not zero due to the Fresnel reflection of the sensing fiber and the presence of system noise, resulting in the phenomena of multiple peaks and pseudo-peaks in the cross-correlation results, which has a great impact on the variable demodulation results.

In order to avoid the shortcomings of the basic cross-correlation algorithm and improve its poor calculation effect in the case of a low signal-to-noise ratio, this paper uses a generalized cross-correlation algorithm based on the weighting function of PHAT to process the local reference signal and the local measurement signal. A flowchart of the generalized cross-correlation algorithm is shown in Figure 4.

(1)The local reference signal and the local measurement signal are Fourier transformed.



(14)
$$\begin{array}{l} F_{s_{r}\left(n\right)}\left(w\right)=\int_{-\infty }^{\infty }s_{r}\left(n\right)e^{-jwn}dn\\ F_{s_{m}\left(n\right)}\left(w\right)=\int_{-\infty }^{\infty }s_{m}\left(n\right)e^{-jwn}dn \end{array}$$



(2)The conjugate of the transformed local measurement signal is taken, and the two signals are convoluted to obtain the cross-correlation power spectrum *G*_12_(*w*) as follows:

(15)$$G_{s_{r}s_{m}}(w)=F_{s_{r}}(w)F_{s_{m}}^{*}(w)$$
where $F_{s_{m}}^{*}(w)$ represents the conjugate complex of $F_{s_{m}}(w)$. According to Venasinchin’s theorem, the Fourier transform of the cross-correlation function is the cross-power spectral density, so the cross-correlation function of the local measurement signal and the local reference signal can be written as
(16)$$R_{s_{r}s_{m}}(k)=\frac{1}{2\pi }\int_{-\infty }^{\infty }G_{s_{r}s_{m}}(w)e^{jwk}dw$$

There are multiple peaks or pseudo-peaks in the calculation result due to the noise between the local measurement Rayleigh signal and the local reference Rayleigh signal.

(3)The weighting function of the cross-power spectrum is calculated:

(17)$$R_{s_{r}s_{m}}^{\left(g\right)}(k)=\frac{1}{2\pi }\int_{-\infty }^{\infty }\psi (w)G_{s_{r}s_{m}}(w)e^{jwk}dw$$
where *ψ*(*w*) represents the weighting function. According to a priori statistical information, PHAT can effectively sharpen the peak value of generalized cross-correlation results.

The generalized cross-correlation algorithm whitens the signal and noise by calculating the cross-power spectrum and giving it a certain weight to enhance the components with high signal-to-noise ratios and to suppress the impact of the noise. Figure 5 shows the results of the measured local Rayleigh scattering signal and the local reference Rayleigh scattering signal with a signal-to-noise ratio of 10 dB. It can be seen in Figure 5a that the generalized cross-correlation-based PHAT can still effectively calculate the correlation coefficient in the case of a low signal-to-noise ratio. In contrast, the traditional cross-correlation operation shows the phenomena of multi-peaks and pseudo-peaks, which affect the actual spectral drift demodulation effect, as shown in Figure 5b.

## 3. Experimental Results

### 3.1. Experiment

The optical path diagram presented in Figure 6 was built based on optical frequency domain reflectometry technology to obtain the Rayleigh scattering spectrum of a sensing fiber. The light source (Yenista Tunics Reference) entered the auxiliary interferometer and the main interferometer from coupler 1 with a tuning range of 1540 nm~1560 nm and a tuning speed of 40 nm/s. The auxiliary interferometer was a Michelson interferometer with a delay fiber length of 20 m. The signal of the main interferometer was resampled at an equal frequency to eliminate the influence of the nonlinear tuning of the tuned light source. The reference light and the measured light in the main interferometer entered OC4, were received by the polarization balance detector (Thorlabs Int-Pol-1550), and were collected by the acquisition card (Advantech PCI-1714). The sensing fiber was a single-mode fiber produced by Corning with a length of 2 m.

In order to apply uniform strain to the sensing fiber, the 0.18 m position of the sensing fiber was glued on the stepper motorized stage (Suruga Seiki KXC06, with a displacement resolution of 50 nm), and the 0.45 m position was fixed on the fiber-holding mechanism, as shown in Figure 7. The axial strain could be applied to the sensing fiber by moving the micro-displacement platform. The Rayleigh scattering spectrum of the sensing fiber was recorded as the reference spectrum when the lateral movement of the micro-displacement table was 0. The sensor fiber was stretched in steps of 500 με, and the Rayleigh scattering spectrum under each strain state was recorded as the measurement spectrum.

### 3.2. Discussion

Figure 8 shows the cross-correlation calculation results of the measurement spectrum and the reference spectrum at the position of 0.18 m~0.45 m of the sensing fiber under different strain conditions. WS represents the traditional strain demodulation algorithm, and LS is the strain demodulation algorithm proposed in this paper. The spatial resolution was set to 10 mm, 5 mm, and 3 mm to verify the influence of the spatial resolution on the calculation results. It can be seen that the correlation coefficient between the measured spectrum and the reference spectrum of the traditional strain demodulation algorithm is reduced to less than 0.2 when the sensing fiber is 500 με, and the problem cannot be solved by increasing the spatial resolution. This is due to the degradation of the measured signal caused by the strain; thus, the similarity with the reference spectrum decreases sharply. However, increasing the spatial resolution only increases the number of cross-correlations and does not increase the similarity between the reference spectrum and the measurement spectrum. The method proposed in this paper still maintains a high spectral similarity when the sensing fiber is subjected to 4000 με. In addition, there is no significant difference in the correlation coefficient when the spatial resolution changes; this is because only the local spectrum is used for demodulation, so the spatial resolution is not the main factor affecting the demodulation results.

To investigate the distributed strain demodulation capability of the proposed method, an axial strain was applied in steps of 500 με with a range of 1500~4000 με to the 0.18~0.45 m segment of the sensing fiber. The measurement results, with a spatial resolution of 3 mm, are shown in Figure 9b. It can be seen that the demodulation result of any strain value does not appear in the error point of the strain measurement caused by multi-peaks or pseudo-peaks, showing that the strain demodulation method proposed in this paper has high stability. Figure 9a shows the fitting curves of the strain and spectral shift, which maintain a highly linear relationship.

In order to study the accuracy of the proposed method for strain measurements, an axial strain of 4000 με was applied to the sensing fiber, and the strain demodulation had a spatial resolution of 3 mm. The strain measurement results of each position were taken as samples (90 sampling points). The standard deviation was calculated to be 6.2 με.

## 4. Conclusions

Firstly, a Rayleigh scattering model based on the refractive index perturbation of an optical fiber is proposed to study the characteristics of Rayleigh scattering and to guide the strain demodulation algorithm based on the spectral shift. Secondly, a local similarity method is proposed, and the algorithm still has a high spectral similarity between the local reference Rayleigh scattering signal and the local measured Rayleigh scattering signal, solving the phenomenon of multi-peaks or pseudo-peaks, which occur due to the reduction in the spectral similarity in the cross-correlation process of traditional algorithms. Thirdly, a generalized cross-correlation algorithm is proposed to solve the problem of demodulation failure caused by a Rayleigh scattering signal with a low signal-to-noise ratio. Experiments show that the proposed method still has high stability when the spatial resolution is 3 mm, and the measurement precision is 6.2 με. The strain measurement technology proposed in this paper based on OFDR can greatly improve the strain measurement range of optical fiber sensors, and it meets the needs of health monitoring in construction and industry, modern medical fields, and other fields.

## Figures and Tables

**Figure 1 sensors-22-05378-f001:**
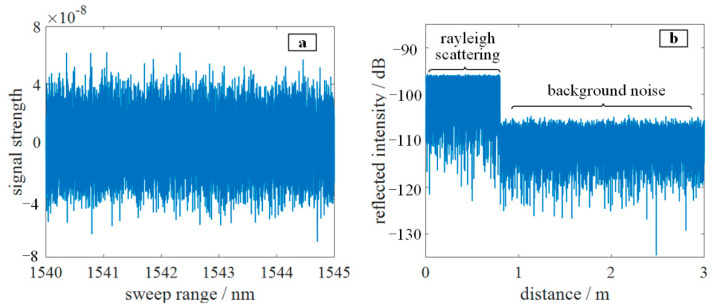
Schematic diagrams of Rayleigh scattering simulation of optical fiber: (**a**) Rayleigh scattering signals in the frequency domain; (**b**) Rayleigh scattering signal in the range domain (Rayleigh spectra).

**Figure 2 sensors-22-05378-f002:**
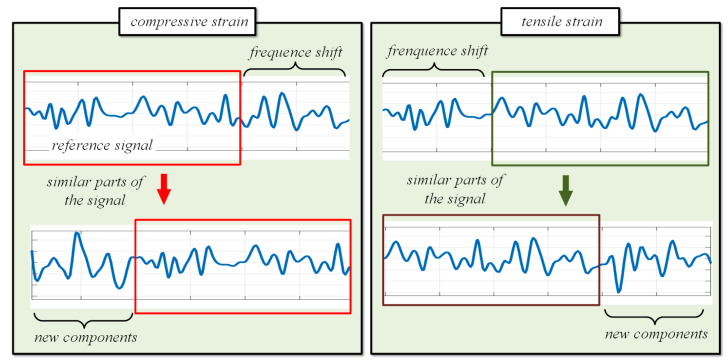
Schematic diagram of spectrum shift. Tensile and compressive strains cause a frequency shift, which can be solved using cross-correlation operation. The new components in the diagram are introduced by the shift. The longer the length of the new component, the shorter the length of the similar parts of the signal, resulting in the failure of the cross-correlation operation.

**Figure 3 sensors-22-05378-f003:**
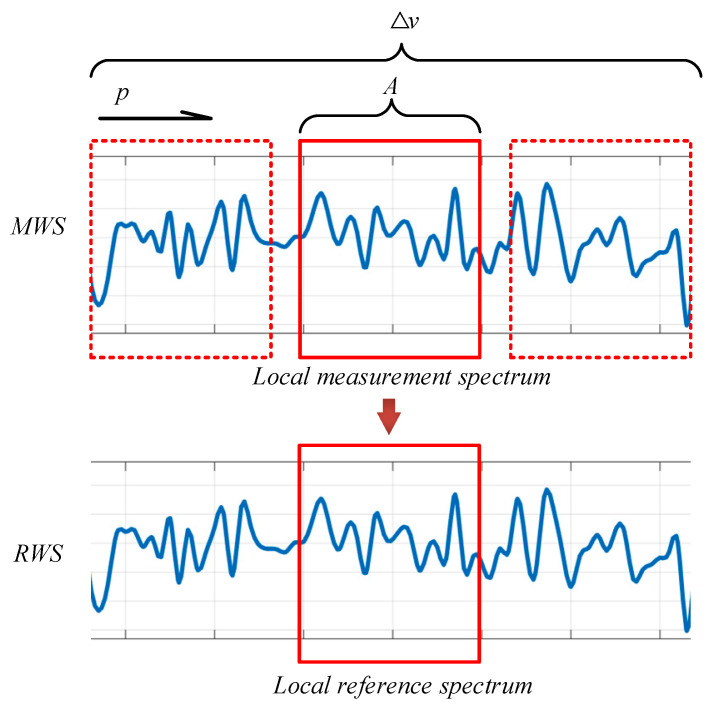
Schematic diagram of demodulation method based on local similarity. The local measured Rayleigh signal (LMS) is obtained by intercepting the measured whole Rayleigh scattering signal (MWS) with a rectangular frame of length A, and its position is represented by *p*. A local reference Rayleigh signal (LRS) of length A is intercepted at the middle of the reference whole Rayleigh scattering signal (RWS). The correlation coefficient between the LRS and the LMS is calculated at each position.

**Figure 4 sensors-22-05378-f004:**
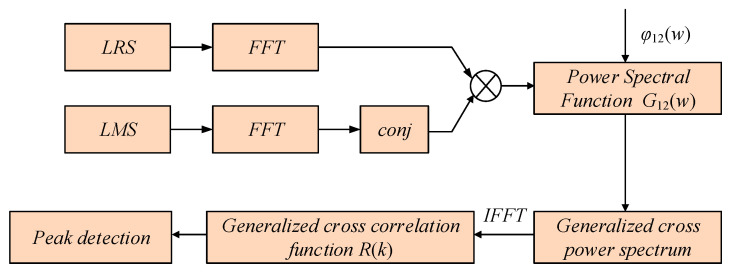
Flowchart of generalized cross-correlation algorithm. The local reference signal and the local measurement signal are Fourier transformed. The conjugate of the transformed local measurement signal is taken, and the two signals are convoluted to obtain the cross-correlation power spectrum. The generalized cross-correlation function is obtained using the inverse Fourier transform of the generalized cross-correlation power spectrum weighted by the cross-correlation power spectrum. The peak position of *R*(*k*) is recorded.

**Figure 5 sensors-22-05378-f005:**
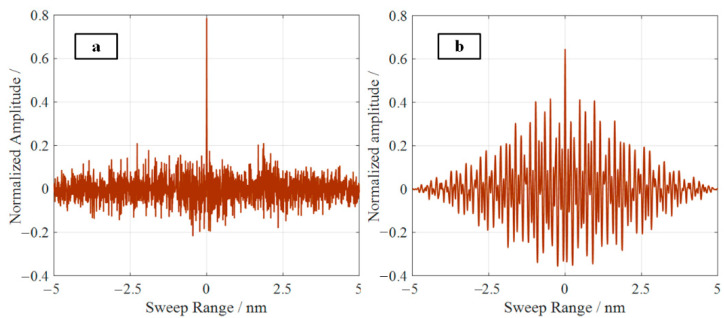
Results of traditional and generalized cross-correlation operations: (**a**) results of the generalized cross-correlation operation based on PHAT; (**b**) results of the traditional cross-correlation operation.

**Figure 6 sensors-22-05378-f006:**
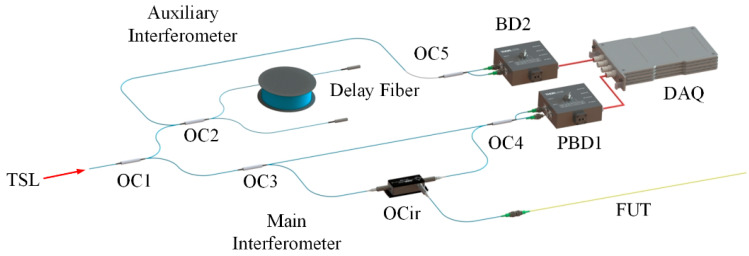
Optical path diagram. TLS—tuning sweep light source; OC—optical coupler; OCir—optical circulator; BD—balance detector; PBD—polarization balance detector; DAQ—data acquisition card; FUT—fiber under text.

**Figure 7 sensors-22-05378-f007:**
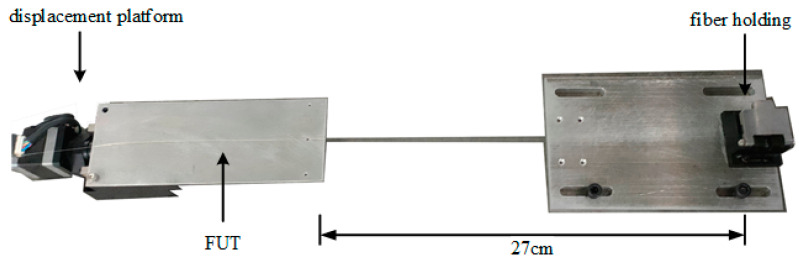
Structure for applying axial strain to fiber under text.

**Figure 8 sensors-22-05378-f008:**
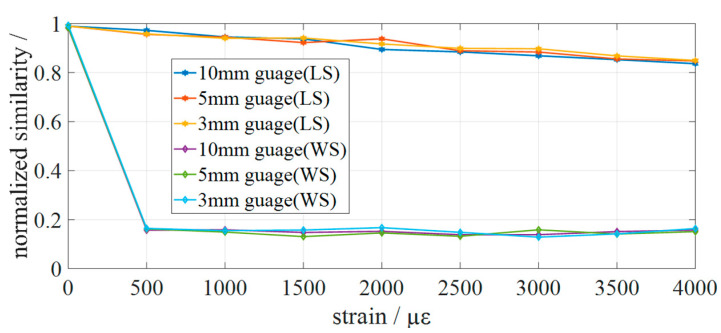
The demodulation results. The correlation coefficient between the measured spectrum and the reference spectrum of the traditional strain demodulation algorithm is reduced to less than 0.2 when the sensing fiber is 500 με. The method proposed still maintains high spectral similarity when the sensing fiber is subjected to 4000 με.

**Figure 9 sensors-22-05378-f009:**
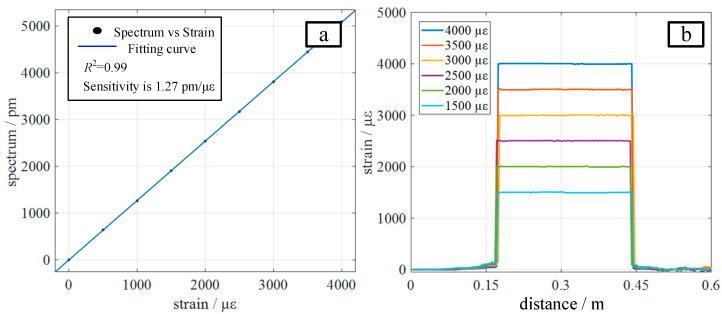
Distributed strain measurement results: (**a**) shows the fitting curve of strain and spectral shift, which maintain a highly linear relationship, and (**b**) shows the distributed strain demodulation result of the proposed method.

## Data Availability

Not applicable.

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
