# Peer review of "Generalized Cross-Correlation Strain Demodulation Method Based on Local Similar Spectral Scanning"

_sensors, 2022, doi:10.3390/s22145378_

Round 1
Reviewer 1 Report
Manuscript No: sensors-1818092
Title: Generalized cross-correlation strain demodulation method 2 based on local similar spectral scanning
Authors: Yuqi Tian, Jiwen cui, Zaibin Xu and Jiubin Tan
A. Overview
1. In this manuscript the authors report on a methodology for cross-correlation strain demodulation based on similar spectral scanning.
2. The contents are expressed clearly.
3. The manuscript could be better organized,
4. It is written in reasonable English.
5. The authors have acknowledged recent related research.
6. As long as my knowledge, the work presented is original.
7. Several typos along de manuscript, authors must have a second read.
B. Detailed analysis.
Abstract
-Please organize the ideas in each paragraph.
-Be clear, objective.
-State briefly what you did, how did you do it, the quantitative results you and
-State clearly the novelty of your work.
1.Introduction: (not 1.Introduce - typo)provides an interesting approach to the subject and there are up to date references.
2.Rayleigh scattering in optical fiber - I advise the creation of a section called 2. Theory: where the relevant analysis theory is presented, subdivided into 3 subsections:
2.1 Rayleigh scattering in optical fiber and 2.2 Local similar scanning algorithm and 2.3 Generalized cross correlation algorithm
5. Experiment and analysis – this should be called Experimental Results and must be divided into Material and Methods and Results and Discussion.
Here a meticulously discussion of the results is missing
Improve the quality of the figures, particularly figures 6 and 7.
C. Overall assessment
The work presented here is very interesting and has potential for further development in the field. In my opinion the manuscript must be checked again after an improvement of the organization of the manuscript and a comprehensive discussion of the results. Also, the novelty of the work must be clearly stated.
D. Review Criteria
1. Scope of Journal
Rating: Medium
2. Novelty and Impact
Rating: Medium
3. Technical Content
Rating: Medium
4. Presentation Quality
Rating: Medium
Reviewer 2 Report
In this manuscript, authors introduced a development of cross-correlation strain demodulation method for optical fiber strain sensors, which have been used in industrial and biomedical fields. I determined that this research manuscript is suitable for a publication in Sensors since it comes to research topics in the journal. On the other hand, there are a lot of points, which should be revised before the publication, in the submitted manuscript. (1) Introduction was not sufficiently presented related studies and the reason why this study is necessary. The section of Introduction should be sufficiently supplemented with an addition of reference research articles related to authors' study. (2) Figure 1 does not include a schematic diagram of the simulation. Authors should include the schematic diagram in the figure or revise a figure caption. (3) A detailed description of Figure 2 and Figure 3 shall be added in a caption of Figure 2 and Figure 3 to enhance a degree of readers' understanding. (4) Similarly, a detailed explanation of each component and sequence for a flow chart in Figure 4 should be included in a caption of Figure 4. (5) (Line 212) "Fi.5 shows the simulation results of the measured ~" This sentence needs to be corrected for typos. (6) Descriptions of an abbreviation of each optical component in Figure 6 and an overview of the developed optical system should be added to a caption of Figure 6. (7) In Fig. 9, authors presented a fitting curve of a relationship between spectrum and strain. An equation and R-square value of the fitting curve must be added in Fig. 9. Also, detailed descriptions of Fig. 8 and Fig. 9 are required before the publication. (8) An additional explanation of where this research result can be applied should be supplemented in a section of Conclusion. A sufficient number of reference articles related to optical fiber sensors will be needed to present it. (9) It is necessary to correct English and to match the presentation of numbers and units. For instance, "Experiments shows that the proposed method still has high stability when the spatial resolution is 3mm, and the measurement precision is 6.2 με." Select a single presentation ((Number) (Unit) or (Number)(Unit)). Also, 'Experiments shows ~' should be revised to 'Experiments show ~'. I recommend you get help from a professional English grammar correction agency.Author Response
Please see the attachment

Round 2
Reviewer 2 Report
I determined that a revised manuscript fully reflects the comments. On the other hand, I think that English grammar in the revised manuscript should be checked one more time. It requires a request from the editor before proofreading.